# Personalized Case- and Evidence-Based TBI Prognosis with Small Language Models

Pranav Manjunath
*Dept. of Biomedical Engineering*
*Duke University*
Durham, USA
pranav.manjunath@duke.edu

Syed M. Adil, MD
*Dept. of Neurosurgery*
*Duke University*
Durham, USA
syed.adil@duke.edu

Benjamin D. Wissel, MD
*Dept. of Neurosurgery*
*Duke University*
Durham, USA
benjamin.wissel@duke.edu

Daniel P. Sexton, MD
*Dept. of Neurosurgery*
*Duke University*
Durham, USA
daniel.p.sexton@duke.edu

Brian Lerner
*Dept. of Electrical and Computer Engineering*
*Duke University*
Durham, USA
brian.lerner@duke.edu

Timothy W. Dunn
*Dept. of Biomedical Engineering*
*Duke University*
Durham, USA
timothy.dunn@duke.edu

*Abstract*—Timely and accurate emergency department disposition for traumatic brain injury patients requires rapid synthesis of complex, multimodal data. Yet in practice, such decisions often rely on heuristics, resulting in variable outcomes. While large language models show promise for supporting evidence-based practice, their clinical deployment is limited by size, cost, and privacy concerns. We present a dual retrieval-augmented framework that leverages efficient, on-premise small language models and unifies evidence-based practice with case-based reasoning to enable personalized disposition prediction of patients with traumatic brain injury. Evidence-based practice is modeled by retrieving guideline passages tailored to each patient's presentation, while case-based reasoning retrieves similar patients as few-shot exemplars. This dual-retrieval strategy personalizes both clinical guidelines and case-based exemplars, enabling the language model to produce predictions that integrate guideline alignment with patient-specific context. We implemented this framework using two open-source language models under 4B parameters—Phi-4-mini and Qwen-2.5. Across both models, similar patient exemplars consistently improved classification performance, increasing sensitivity without sacrificing specificity. Clinical guidelines had less impact on performance, but when combined with exemplars, they shifted predictions toward more conservative, guideline-consistent behavior. Clinician evaluations suggest that while adding similar patient exemplars enhances accuracy, overreliance on exemplars may diminish reasoning quality, whereas guidelines improve the clinical relevance and justification of model outputs. These findings underscore how targeted retrieval can personalize both predictions and their rationale, enhancing the performance, interpretability, and trustworthiness of AI-assisted clinical decision-making.

*Index Terms*—Small Language Models, Case-based Reasoning, Evidence-based Practice, Traumatic Brain Injury, ED Disposition

## I. INTRODUCTION

Traumatic Brain Injury (TBI) is a complex and heterogeneous clinical condition that manifests in a broad spectrum of presentations in the emergency departments (EDs). Making timely and accurate disposition decisions (e.g., admit to intensive care, admit to floor, or discharge) is critical in such cases, yet often challenging: clinicians must synthesize multimodal evidence—including radiology findings, physical exams, laboratory results, vital signs, clinical history, and formal practice guidelines—under intense time pressure [1]. ED disposition serves as an initial prognosis, directly informing resource allocation and guiding downstream care pathways. However, existing workflows rely heavily on individual clinician experience and heuristics, yielding variability in decision quality and outcomes [2], [3].

Evidence-based practice (EBP) is a core principle in clinical care, defined by the integration of individual clinical expertise with the best available external evidence from systematic research [4]. It emphasizes making patient care decisions that are conscientious, explicit, and grounded in the most current and reliable clinical research. However, EBP alone can be insufficient in high-stakes, data-rich environments like the ED, where patients often present with atypical or ambiguous findings [5].

To address these limitations, case-based reasoning (CBR) offers a complementary strategy by drawing on prior patient cases with similar presentations and outcomes to inform current decisions [6]. CBR enables clinicians to incorporate real-world precedent into their reasoning, supporting personalization in complex or uncertain cases. By combining the population-level rigor of EBP with the individualized relevance of CBR, clinicians can arrive at decisions that are both evidence-informed and context-aware.

Recent studies have explored the potential of large language models (LLMs) in encoding clinical knowledge [7] and augmenting EBP [8], [9], highlighting their strong zero-shot reasoning capabilities, effective summarization, and knowledge transfer through prompting. Kresevic S. et al. [10] demonstrated the impact of retrieval-augmented generation (RAG) and prompt engineering in enhancing clinical decision support, achieving near-perfect accuracy in chronic infection management through structured guideline integration. In emergency

triage, RAG augmented, but not standalone [11], LLMs have outperformed emergency medical technicians and physicians in time-critical decision-making [12]. Beyond generating predictions, LLMs can also provide clinically relevant reasoning, offering justifications that reflect underlying clinical logic and support more transparent decision-making [13], [14].

Despite their capabilities, LLMs face practical barriers in clinical deployment due to their size, cloud-based inference, and the risk of exposing protected health information [15]. Small language models (SLMs) present a promising alternative: they are lightweight, open-source, and can be securely deployed behind hospital firewalls, enabling privacy-preserving, real-time decision support. Yet, most evaluations of SLMs have focused on standardized medical licensing examinations (e.g., USMLE) [16], which do not reflect the multimodal complexity or time-sensitive nature of real-world ED practice.

In this work, we present a dual-retrieval framework to evaluate how integrating EBP and CBR influences the performance and reasoning of SLMs for personalized clinical prediction of ED disposition. Our system retrieves both relevant clinical guidelines to model EBP and similar patients as few-shot exemplars for CBR, enabling a controlled assessment of each strategy, and their combination, on prediction accuracy and clinical reasoning. We further personalize EBP by selecting only the most relevant content for a patient's presentation, allowing the model to contextualize recommendations while remaining aligned with best practices. All predictions are generated purely via in-context learning, without any fine-tuning. Our approach advances toward an end-to-end precision health system that is evidence-aligned, personalized, and suited for high-stakes settings like TBI care in the ED. Our key contributions are:

1) A personalized retrieval-augmented framework that unifies EBP and CBR within a privacy-preserving SLM. The framework first retrieves the most relevant guideline passages for a test patient, then augments the prompt with similar patients from multimodal embeddings, enabling predictions and reasoning that are both guideline-aligned and case-informed.
2) A systematic evaluation of retrieval-augmented strategies using clinical guidelines to predict ED disposition in patients with TBI.
3) Investigation of how the clinical reasoning of SLMs shifts under different prompting strategies and levels of information granularity, evaluated in collaboration with clinicians.

## II. DATA

We utilized a multimodal dataset from the RAPID-TBI study [17], a comprehensive resource for adult TBI prognosis and emergency triage. We focused on a held-out test set comprising of 1,411 ED encounters from two hospitals within the Duke University Health System. Throughout this paper, we use the term patient to denote an encounter, as each encounter is treated independently. This test set included adult patients with TBI (mean age±s.d.: 59.05 ± 21.31 years; 54.8%

female). The cohort was predominantly composed of mild TBI cases, with 81.4% of patients presenting with an initial Glasgow Coma Scale (GCS) score of 15. The ED disposition outcome was binarized into "Discharge Home" and "Admit to Hospital" (including floor, ICU, or surgery), with 68.8% of patients discharged home.

Each patient record includes a rich combination of structured and unstructured clinical data recorded in the ED: (i) radiology report impressions from initial head CT scans (ii) patient details (e.g, age, height, weight, reason for ED visit) (iii) Vitals (e.g, blood pressure, heart rate, respiratory rate, temperature) (iv) laboratory analytes (e.g, coagulation, sodium, glucose, hematocrit, platelets) (v) other diagnostic orders requested during the ED stay. Since many of these features are recorded multiple times throughout the ED course, we summarized them using the arithmetic mean, measurement count (number of times recorded), and range (minimum, maximum). To enable evaluation with SLMs, we converted each multimodal patient record into a structured, text-based patient presentation shown in Fig. 2. This representation captures the key clinical context available to ED clinicians and structures it in a format readily interpretable by language models.

## III. METHODS

### A. CBR Similar Patient Retrieval

For each test patient, we retrieved the Top-$n$ most similar patients from an independent retrieval database of 6,856 patients drawn from the same population. We used RAPID-TBI [17] to encode multimodal patient data, including imaging, vitals, labs, and demographics into a unified embedding, using Facebook AI Similarity Search (FAISS) [18] with cosine similarity to identify the nearest patients in the embedding space. Once retrieved, both the test patient data and the $n$ similar patient exemplars are converted into text (Fig. 2) and appended to the model prompt. Each retrieved exemplar includes its ED disposition, providing real-world cases that serve as in-context personalized examples, a retrieval-based augmentation paralleling few-shot prompting [19], to guide the model's prediction.

### B. EBP Clinical Guideline Retrieval

To construct a RAG pipeline over formal TBI practice guidelines, we assembled a curated database of high-quality clinical guidelines. We selected six guidelines [20]–[25] that scored above 50 across all six domains of the AGREE II instrument [26]. Non-text elements like figures and tables were excluded. The remaining text was segmented into chunks, with each chunk corresponding to a single page column (roughly 200 words). These chunks were indexed using a TF-IDF vectorizer [27] and stored in a searchable database to support efficient, interpretable retrieval during prompting. The guideline database for EBP consisted of a total of 664 chunks.

Our approach for focused retrieval from the guideline database, begins with prompting the SLM to extract the top 3–5 key clinical findings from the test patient's data that are most informative for predicting ED disposition. The SLM

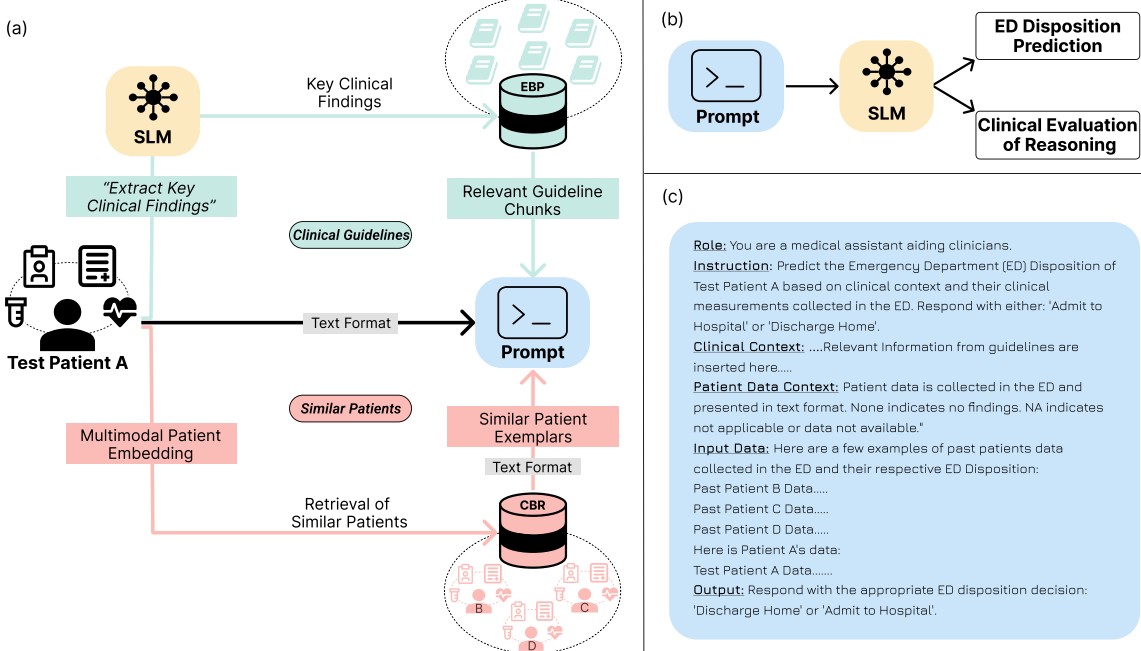

Fig. 1. Overview of the SLM system. (a) Prompts are constructed using retrieved clinical guidelines (EBP) and similar patient exemplars (CBR). (b) The prompt is sent to the SLM for two tasks: (i) ED disposition classification and (ii) structured clinical reasoning generation. (c) Example prompt (without CoT) showing how patient data, guideline evidence, and exemplar cases are combined to guide the response.

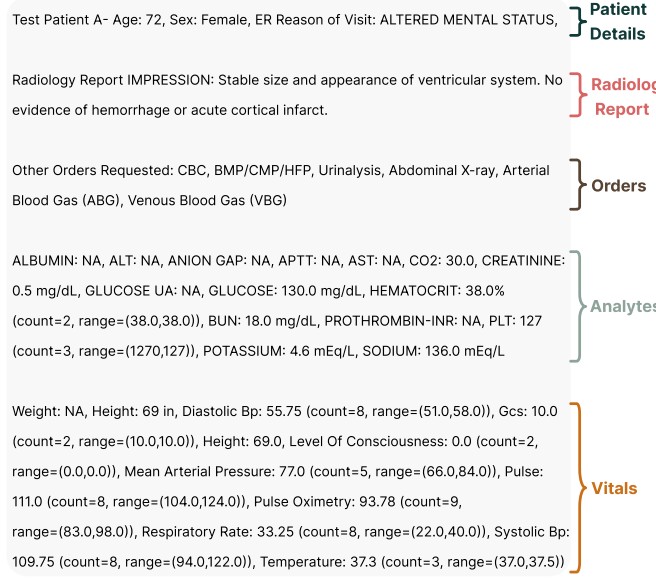

Fig. 2. Test Patient Data Text Format

outputs a structured response consisting of 3–5 sentences, where each sentence encapsulates a distinct clinical observation relevant to disposition decisions (e.g., "Patient A's Glasgow Coma Scale score is 10, indicating a moderate TBI."). These key findings are then vectorized and used to query the clinical guideline chunks. We investigated three retrieval strategies for identifying relevant guideline content, as outlined below:

1) Findings-Global Top-$k$: The entire key findings response $R$ is encoded using TF-IDF to obtain a single vector representation. Then, the cosine similarity is computed between $R$ and each chunk $c_i$ in the corpus. The Top-$k$ chunks with the highest similarity scores are returned.

2) Sentence-Global Top-$k$ Retrieval: The key findings response $R$ is split into individual sentences $\{s_1, ..., s_M\}$, where $M$ is the total number of sentences. Each sentence is encoded with TF-IDF and compared against all guideline chunks using cosine similarity. For every sentence, the Top-3 most similar chunks are retrieved. The union of all retrieved chunks is subsequently aggregated, deduplicated, and sorted by cosine similarity, after which the Top-$k$ globally ranked chunks are returned.

3) Sentence-Local Top-$k$ Retrieval: Similar to the global version, the key findings response is split into sentences, with each sentence encoded using TF-IDF. For every sentence $s_j$ in $\{s_1, ..., s_M\}$, the Top-$k$ most similar chunks are retrieved. The final output consists of a set of sentence-specific Top-$k$ chunks without global ranking across sentences.

We included Sentence-Global and Sentence-Local to facilitate fine-grained alignment between patient features and guideline chunks, as this may be lost with Findings-Global. To assess retrieval quality, we defined Lexical Coverage (Eq. 1), the proportion of meaningful words from key clinical findings ($W_{\text{findings}}$) that also appear in the retrieved guideline text after removing stopwords ($W_{\text{guideline}}$).

$$\text{Lexical Coverage} = \frac{|W_{\text{findings}} \cap W_{\text{guideline}}|}{|W_{\text{findings}}|} \qquad (1)$$

## C. SLM Response Evaluation

In addition to predicting ED disposition, we collaborated with clinicians to evaluate the quality of reasoning generated by the SLM. These responses were independently assessed by three neurosurgery residents (PGY-7, PGY-4, PGY-2), providing diverse clinical perspectives on model-generated outputs. We systematically explored how the SLM's reasoning varied under two prompting comparisons:

1) CBR vs. No CBR: We evaluated how the presence of CBR affected SLM response by comparing three prompting strategies: zero-shot, few-shot with clinically similar patients (CBR), and few-shot with randomly selected patients. This comparison was designed to isolate the specific contribution of clinically meaningful exemplars to the model's reasoning quality. In this experiment, each clinician reviewed 20 patient cases, with each case comprising four responses (zero-shot, 1-shot, 5-shot, and 5-shot random), for a total of 80 responses per clinician. The responses were presented in random order. A total of 30 unique cases were evaluated, with each case independently reviewed by two clinicians.

2) CBR vs. CBR + EBP: We assessed whether augmenting CBR with EBP improved the model's reasoning quality. In this experiment, a clinician reviewed 30 patient cases, with each case comprising two responses: one using CBR alone and one combining CBR with EBP.

To structure this evaluation, we adopted the QUEST framework [28], a clinically grounded rubric for assessing medical reasoning in language model outputs. Table I summarizes the key QUEST criteria we employed. This framework enabled a principled and expert-aligned qualitative analysis of the model's decision-making rationale. All metrics were scored using a 5-point Likert Scale, where 5 indicates the top score.

## D. Experimental Design

We used two SLMs: Phi-4-mini-instruct (3.8B) [29], referred to as Phi, and Qwen-2.5-3B [30], referred to as Qwen. All experiments were run with temperature 0 on an NVIDIA RTX 6000 Ada GPU, to ensure deterministic output. Models were locally downloaded from Hugging Face, loaded via PyTorch, and hosted on a local inference server. For ED disposition prediction, the SLM was prompted to either (i) directly output "Admit to Hospital" or "Discharge Home," or (ii) generate a chain-of-thought (CoT) explanation followed by a prediction. CoT experiments used vLLM [31] for efficient batching. For reasoning evaluations, the SLM produced structured outputs with: (1) key clinical findings, (2) comparison to similar patients (if applicable), (3) clinical reasoning, and (4) the final prediction. For prediction performance metrics, we used: F1 for overall balance, sensitivity (SN) for detecting positives, and specificity (SP) for detecting negatives. Code available: https://github.com/PranavM98/TBI-Prognosis-with-SLM

## IV. RESULTS AND DISCUSSIONS

### A. SLM Prediction Evaluation

All evaluations used the full test set of 1,411 patients unless stated otherwise. Consistent with prior work [32], CoT reasoning did not improve ED disposition prediction, with or without CBR (Table II). Due to poor CoT output quality, 23 test cases were excluded from this specific experiment. Based on these results, we focus exclusively on non-CoT prompting in the remainder of the this section.

Table III presents the overall performance of different prompting strategies evaluated on both SLMs using F1, SN, SP as metrics. A key observation across both models is that CBR consistently improved performance relative to both the zero-shot baseline (without both EBP and CBR) and the EBP strategy (Sentence-Local). The highest F1 scores for both models were achieved with CBR alone, specifically with $n = 5$ or $n = 7$, highlighting the value of providing context through similar retrieved patient exemplars. Interestingly, while adding EBP to CBR did not yield significant additional gains in F1, adding CBR to an EBP-only setup consistently enhances performance—underscoring the stronger contribution of patient-specific exemplars over guideline retrieval in improving model predictions. Random selection of patients ($n = 7$), resulted in notably lower performance (F1), emphasizing the importance of exemplar relevance.

A closer look at SN and SP tradeoffs in Table III reveals that EBP (Sentence-Local) dramatically boosted sensitivity but at the cost of a sharp decline in specificity. This trend was pronounced across both SLMs. EBP (Top-3) increased SN to 0.948 but reduced SP to just 0.202 for Phi. Qwen reached its highest SN (0.957) with EBP Top-1, but with a corresponding drop in SP to 0.238. In contrast, CBR achieved a more stable balance between SN and SP. For instance, CBR with five similar patients in Phi yielded 0.853 SN and 0.601 SP, and in Qwen 0.707 SN and 0.792 SP.

Notably, the effect of EBP differed between the two models. For Phi, incorporating EBP alone actually lowered the F1 score compared to the zero-shot baseline, suggesting that exposure to guidelines may introduce conflicting signals. In contrast, Qwen saw modest F1 gains with EBP alone, although its baseline performance was substantially lower. When combining EBP with CBR, Phi achieved a more balanced prediction profile (0.698 SN, 0.758 SP), while Qwen remained skewed toward sensitivity (0.912 SN, 0.399 SP). These performance trends were generally consistent across TBI severity levels (Table IV). However, we found that Phi F1 score improved for moderate TBI patients when incorporating Sentence-local CBR. These results motivate our focus on Phi for ablation studies across different RAG strategies.

When comparing RAG strategies (Table V) using only EBP (CBR: $n = 0$), we observed that Sentence-Global yielded the highest F1 score (0.548), followed closely by Sentence-Local (0.544), with Findings-Global performing the worst (F1 = 0.531). This suggests that breaking down the key clinical findings into individual sentences and retrieving relevant chunks

TABLE I
*Clinician evaluation framework for assessing LLM-generated ED disposition reasoning.*

| Principle | Dimension | LLM Evaluation Metric | Definition |
|---|---|---|---|
| **Quality of Information** | Accuracy | Fact Recapitulation | Does the AI accurately summarize and retrieve the patient's clinical details? |
| | Accuracy/ Agreement | Fact Labeling | Are the clinical facts stated correctly? |
| **Understanding and Reasoning** | Reasoning | Clinical Reasoning | Given the facts presented (whether accurate or not), is the downstream reasoning logical and medically appropriate? |
| **Safety and Harm** | Bias | Bias | Is there evidence of systematic bias, such as inappropriate use of gender? |
| | | Overall Clinical Judgement | As an overall assessment, how do you feel the AI did in distilling the relevant facts and coming to a clinically sound decision regarding ED disposition? |

TABLE II
*F1 scores for ED disposition using Phi with and without CoT reasoning across 1,388 test patients.*

| | $n=0$ | $n=1$ | $n=3$ | $n=5$ | $n=7$ |
|---|---|---|---|---|---|
| Without CoT | 0.585 | 0.547 | 0.587 | 0.626 | 0.635 |
| With CoT | 0.505 | 0.560 | 0.570 | 0.494 | 0.522 |

TABLE III
*Overall predictive performance across models: EBP reported reflect the Sentence-Local strategy, while the combined EBP and CBR strategy includes EBP Top-3 with CBR using $n=5$*

| Prompting Strategies | Phi | | | Qwen | | |
|---|---|---|---|---|---|---|
| | **F1** | **SN** | **SP** | **F1** | **SN** | **SP** |
| **Zero Shot** | 0.586 | 0.728 | 0.656 | 0.386 | 0.311 | 0.865 |
| **EBP Top-1** | 0.544 | 0.821 | 0.455 | 0.527 | **0.957** | 0.238 |
| **EBP Top-2** | 0.525 | 0.901 | 0.293 | 0.539 | 0.930 | 0.310 |
| **EBP Top-3** | 0.512 | **0.948** | 0.202 | 0.547 | 0.923 | 0.340 |
| **CBR** ($n=3$) | 0.585 | 0.782 | 0.594 | 0.600 | 0.553 | **0.868** |
| **CBR** ($n=5$) | 0.625 | 0.853 | 0.601 | **0.653** | 0.707 | 0.792 |
| **CBR** ($n=7$) | **0.634** | 0.884 | 0.589 | 0.626 | 0.683 | 0.774 |
| **Random** ($n=7$) | 0.540 | 0.887 | 0.365 | 0.565 | 0.644 | 0.711 |
| **EBP and CBR** | 0.626 | 0.698 | **0.758** | 0.563 | 0.912 | 0.399 |

TABLE V
*Prediction Performance: RAG Strategies for Phi. For each strategy, we report the highest F1 score achieved both with and without incorporating CBR.*

| RAG Strategy | CBR Top-$n$ $n$ | EBP Top-$k$ $k$ | **F1** | **SN** | **SP** |
|---|---|---|---|---|---|
| **Findings-Global** | 0 | 5 | 0.531 | 0.553 | 0.759 |
| | 3 | 5 | 0.579 | 0.488 | 0.910 |
| **Sentence-Global** | 0 | 5 | 0.548 | 0.580 | 0.756 |
| | 7 | 7 | 0.595 | 0.500 | **0.919** |
| **Sentence-Local** | 0 | 1 | 0.544 | **0.821** | 0.455 |
| | 5 | 3 | **0.626** | 0.698 | 0.758 |

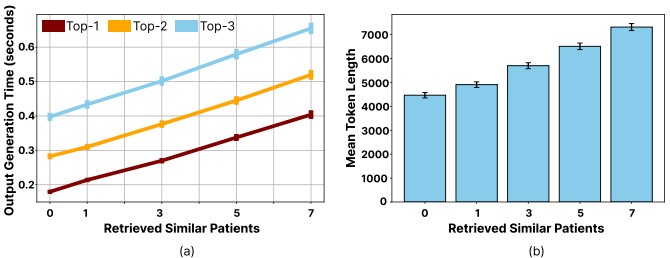

Fig. 3. Results on 200 cases using Phi for ED disposition prediction (Error bars indicate the 95% CI). (a) Output generation time for Sentence-Local EBP (Top-$k$) with CBR at varying numbers of retrieved patients. (b) Mean input token length for Sentence-Local EBP (Top-3) across retrieved patient counts.

per sentence leads to better prediction performance than using the full prompt as a single query. The best CBR and EBP combined performance was achieved with the CBR ($n=5$), EBP (Sentence-Local Top-3), resulting in the highest F1 score of 0.626. Importantly, this configuration also maintained a strong balance between SN (0.698) and SP (0.758), unlike other strategies, which tended to skew toward one at the expense of the other. Based on these findings, we focus on the Sentence-Local strategy for further analysis. For the best F1 Sentence-Local strategy, Lexical Coverage was 0.621 (s.d: 0.079), and for Sentence-Global 0.769 (s.d: 0.069).

These results highlight the complementary yet competing roles of CBR and EBP in SLM-based decision support. CBR boosts F1 and sensitivity by leveraging analogical reasoning from prior cases, capturing more true admissions. EBP, in contrast, increases precision but lowers sensitivity, reflecting a conservative bias toward discharge unless guideline-backed evidence supports admission. This improves specificity and standardization but risks undertriaging borderline cases. Balancing CBR and EBP is thus essential for both safety and adherence to clinical practice in high-stakes decisions.

We evaluated system efficiency by measuring execution time and input token length. Execution time was decomposed into: (i) EBP retrieval (Sentence-Local Top-3), averaging 2.713 seconds; (ii) CBR retrieval ($n=7$), averaging 0.108 ms; and (iii) output generation of ED disposition prediction, with the best-performing dual prompt (CBR: $n=5$, EBP: Top-3) averaging 0.579 seconds (Fig. 3a). The corresponding mean input token length was 6,510 (Fig. 3b).

TABLE IV
*EBP Sentence-local (Top-3) + CBR ($n=5$) across various TBI Severity Levels. $N_{test}$ indicates the number of test patients*

| TBI Severity | SLM | **F1** | **SN** | **SP** |
|---|---|---|---|---|
| **Mild (GCS 13-15)** | Phi | **0.574** | 0.656 | 0.761 |
| $N_{test}$ = 1315 | Qwen | 0.518 | 0.895 | 0.406 |
| **Moderate (GCS 9-12)** | Phi | **0.903** | 0.875 | 0.769 |
| $N_{test}$ = 61 | Qwen | 0.879 | 0.979 | 0.077 |
| **Severe (GCS 3-8)** | Phi | 0.903 | 0.933 | 0.2 |
| $N_{test}$ = 35 | Qwen | **0.923** | 1 | 0 |

## B. LLM Response Evaluation

Fig. 4 shows how varying the number of CBR exemplars influences clinician-rated reasoning quality for Phi, highlighting the isolated impact of patient similarity without guideline-based prompting. Although providing five similar patients ($n = 5$) yielded the highest predictive performance, clinicians rated these responses lower than those generated with no CBR or with just one similar patient across all four evaluation metrics. Surprisingly, even having randomly selected patients as CBR led to higher judgment scores than five similar ones. Closer analysis suggests that with multiple similar patients, the model tends to overfit, forcing analogies even when clinical features diverge, while with random patients, it more often acknowledges the lack of overlap and avoids misleading reasoning. Though not statistically significant, this trend suggests a trade-off: while CBR can boost accuracy, excessive reliance on superficially similar cases may compromise clinical judgment and interpretability. Fig. 5 compares clinician ratings between CBR alone ($n = 5$) and CBR combined with EBP via the Sentence-Local Top-3 strategy. While predictive performance saw only minor gains with EBP, clinician ratings show consistent—though not statistically significant—improvements in clinical relevance, reasoning, and overall judgment. This suggests that while CBR anchors the model in prediction, EBP helps ground its reasoning in evidence-based standards, leading to more clinically meaningful responses. All responses received the lowest bias score, reflecting appropriate and consistent use of gender.

## V. FUTURE WORK AND LIMITATIONS

This work shows how retrieval-augmented small language models have the potential to advance a more personalized health system. By synthesizing individualized information from both clinical guidelines and real-world patient exemplars, this approach aims moves beyond the one-size-fits-all care and offers adaptive, context-aware clinical support. It is designed to serve as a clinical copilot, distilling the most relevant information for each patient to support timely, informed decision-making. The value of this approach extends beyond clinicians and can also support patients and caregivers. By providing clearer, guideline-informed explanations alongside relatable case examples, such systems can make complex prognosis and treatment options more accessible and easier to understand. Looking ahead, future work will involve user studies to better understand how such retrieval systems can be effectively implemented in clinical practice.

This study has several limitations. First, the human evaluation was limited in scale and clinician diversity, which we plan to address in future work. Second, while the dataset included key clinical features, it did not fully capture the complexity of real-world trauma cases (e.g., polytrauma, presenting complaints). Third, model behavior varied across SLMs, particularly with Qwen, underscoring the need to further test additional architectures and training paradigms in the future. Fourth, we explored only one encoding method (TF-IDF) for EBP and alternative encoding methods should be evaluated.

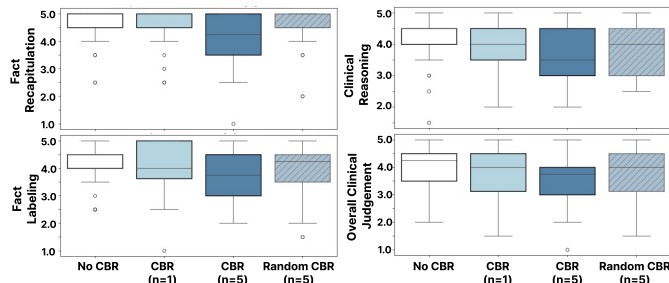

Fig. 4. Phi reasoning quality (QUEST ratings across four dimensions) under different CBR prompt settings.

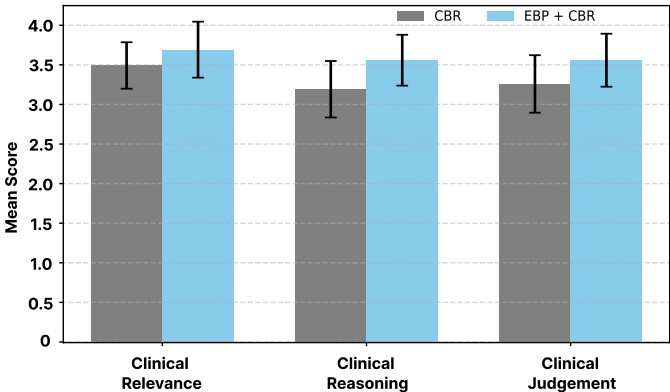

Fig. 5. Box and whisker plots of Phi reasoning evaluation with CBR ($n$=5) with and without EBP. Error bars indicate the 95% CI.

Finally, all analyses were conducted on two SLMs using a single-institution dataset; future work should validate this approach across institutions and with larger models. Future clinical implementation of our approach will require rigorous additional testing of interpretability and robustness across a diverse set of institutions.

## VI. CONCLUSION

In this study, we developed and evaluated a dual retrieval-augmented framework that integrates CBR and EBP, demonstrating its impact on both the performance and reasoning quality of SLMs for predicting ED disposition in patients with TBI. Quantitatively, CBR, by leveraging similar patient exemplars, consistently improved performance and sensitivity across both Phi and Qwen, while EBP had mixed effects, often boosting sensitivity at the cost of specificity. Combining CBR and EBP produced more balanced performance in Phi, though benefits were less consistent in Qwen. Clinician evaluations indicated that CBR can enhance predictions but risks overconfident or forced analogical reasoning when too many similar patients are provided, whereas EBP can improve perceived clinical relevance, reasoning, and judgment. Overall, our study suggests that CBR provides inductive strength, EBP offers deductive grounding, and their combination can improve both accuracy and interpretability. However, SLMs also inherit biases from training data that persist even when given clinically relevant context; in-context learning must balance

personalization with evidence alignment and bias mitigation. Our findings underscore the promise of integrating patient-specific and evidence-guided prompting to enhance the clinical utility of SLM-based decision support in ED care.

## ACKNOWLEDGMENT

This research was supported by an National Institutes of Health grant [R01NS123275]. Our work was approved by the Duke Health Institutional Review Board (Protocol Number: Pro00103826).

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
