# OpenReview forum: "Personalized Case- and Evidence-Based TBI Prognosis with Small Language Models"
_IEEE.org/EMBS/BHI/2025/Conference — BHI 2025_

### Official Review · Reviewer_8J15 · 2025-07-08
**Personalized Case- and Evidence-Based TBI  Prognosis with Small Language Models - Review**

**Confidence:** 5
**Clarity Of Writing:** excellent
**Clinical Significance:** excellent
**Methodological Novelty:** excellent
**Overall Rating:** 8

**Experiments And Results:**

excellent

**Questions For The Authors:**

How do you plan to validate this framework across different hospital systems or more diverse TBI datasets to ensure generalizability?

**Strengths:**

[1] The paper introduces a novel hybrid retrieval-augmented system that combines Evidence-Based Practice (EBP) and Case-Based Reasoning (CBR) with small language models (SLMs), enabling personalized and context-aware clinical predictions.
[2] The authors deployed open-source SLMs (Phi-4-mini and Qwen-2.5), the framework supports real-time, privacy-preserving inference, making it suitable for clinical environments where large models are impractical due to cost and data security concerns.
[3] The combination of CBR and EBP yields more conservative and consistent predictions, striking a balance between sensitivity and specificity that is crucial for high-stakes decisions, such as hospital admissions.
[4] They determine the model’s reasoning capability by using clinicians who use the QUEST framework, rating outputs on accuracy, reasoning quality, and clinical judgment using a 5-point scale. Two structured comparisons: CBR vs. no CBR, and then CBR vs. CBR+EBP, show that while CBR improved accuracy, adding EBP enhanced the clinical relevance and interpretability of model responses.

**Summary Of The Paper:**

The paper introduces a hybrid retrieval-augmented framework that integrates Evidence-Based Practice (EBP) and Case-Based Reasoning (CBR) using small language models (SLMs), Phi-4-mini and Qwen-2.5 under 4B parameters, to predict emergency department (ED) disposition for traumatic brain injury (TBI) patients. By retrieving guideline passages and clinically similar past cases, the system generates personalized and guideline-aligned predictions without requiring fine-tuning. Experimental results show that CBR consistently improves classification performance and sensitivity, while EBP tends to increase sensitivity at the expense of specificity. When combined, EBP and CBR yield more balanced performance in some models and improve the perceived clinical relevance and reasoning quality according to clinician evaluations. The findings highlight that while CBR enhances inductive accuracy, EBP provides deductive grounding, and together they can boost both trustworthiness and effectiveness of AI-assisted clinical decision-making.

**Weaknesses:**

The clinician evaluation component, while valuable, is limited in scale and diversity (three neurosurgery residents). These may not fully capture the range of clinical perspectives or ensure statistical significance in the ratings.

---

### Official Review · Reviewer_vJ7b · 2025-07-09
**Personalized Case- and Evidence-Based TBI Prognosis with Small Language Models**

**Confidence:** 3
**Clarity Of Writing:** great
**Clinical Significance:** great
**Methodological Novelty:** good
**Overall Rating:** 7
**Final Rating:** 8

**Experiments And Results:**

fair

**Questions For The Authors:**

From what I understand, the study uses 1,411 test subjects. When retrieving similar subjects for a given case, are these retrieved subjects part of the same set of 1,411 individuals, or are they drawn from a separate set?

Choosing a temperature of 0 is quite unusual. Is there a specific reason behind this choice? Given that temperature is a key factor influencing text generation diversity, did the authors experiment with different temperature values? Additionally, since language models can produce varying outputs across different random seeds, did the authors evaluate the stability of their results by running multiple trials with different random states?

How much time does it take to compute the similarity and retrieve results? How large the retrieval space was? While the authors argue that their approach is computationally efficient, it's important to note that retrieval also incurs computational cost, especially as the size of the retrieval database grows.

Please provide t the size of the EBP guidelines data used for retrieval

**Strengths:**

The paper leveraged state-of-the-art techniques and integrated to address real-world use cases.

They evaluated by clinicians that is critical aspect when applying AI in healthcare.

**Summary Of The Paper:**

The paper proposes a RAG and context-based approach for leveraging small language models in TBI prognosis. Overall, it presents an innovative and clinically significant contribution to healthcare. However few question needs to be addressed.

**Weaknesses:**

The results do not provide sufficient evidence that the EBP module meaningfully improves model performance, which may be due to the retrieval method. The use of TF-IDF followed by chunk-level similarity may lead to a loss of contextual information from the full document. It would also strengthen the paper if the authors reported the distribution of similarity scores for the retrieved documents, as this would give readers a clearer understanding of how relevant or informative the retrieved content actually is.

The robustness of the proposed method remains unclear. Since the approach involves text generation, it is important to assess how consistent and stable the outputs are across different runs. However, the authors did not report any experiments or analyses that evaluate the robustness of the generated results. It would be helpful if they clarified how robustness was assessed, if it wasn't, consider including such an evaluation to strengthen the claims

---

### Official Review · Reviewer_f73B · 2025-07-12
**Personalized Case- and Evidence-Based TBI Prognosis with Small Language Models**

**Confidence:** 5
**Clarity Of Writing:** good
**Clinical Significance:** fair
**Methodological Novelty:** good
**Overall Rating:** 4
**Final Rating:** 5

**Experiments And Results:**

fair

**Questions For The Authors:**

1. In what ways does the use of the SLM match directly with the decisions that physicians make using the same patient cases?
2. Given the mixed results with EBP, are there specific patient characteristics or clinical scenarios where EBP is more beneficial? How would clinicians determine when to rely on EBP-augmented predictions versus CBR alone?
3. The substantial differences in behavior between Phi and Qwen models raise questions about deployment reliability. How would you address this variability in a clinical setting, and what quality assurance measures would you recommend?
4. Since the data in this dataset may be skewed towards samples of mild TBI, how do you think the system would perform on more severe TBI cases that are more complicated and crucial in terms of disposition? Have you ever thought of validation on a more reasonable distribution of severity?

**Strengths:**

1. The paper demonstrates some real clinical applicability in the manner it solves a real-world critical issue in the emergency medicine field, where all the decisions made concerning dispositions positively affect patient outcomes and resource utilization.
2. The concept of the hybrid EBP+CBR framework is the new integration of clinical guidelines and patient precedents in privacy-friendly SLMs, covering the practical deployment issues of healthcare.
3.  The analysis uses quantitative measures of the performance and qualitative clinician evaluation, making the study comprehensive in its assessment of the system in terms of clinical utility.
4.  The on-premise SLMs tackle the most urgent PHI issues regarding clinical artificial intelligence systems.
5. The paper carefully isolates the contributions of EBP and CBR components by way of controlled comparisons.
6.  RAPID-TBI dataset usage provides realistic clinical data for evaluation.

**Summary Of The Paper:**

This paper proposed a retrieval-augmented framework as a hybrid of the evidence-based practice (EBP) and the case-based reasoning (CBR) approaches, specialized on small language models ( SLMs ) that effectively integrate with emergency department (ED) disposition estimation with traumatic brain injury (TBI) patients. The framework retrieves applicable clinical guidelines (EBP) and analogous cases of patients (CBR), thus offering an individualized context to SLMs. The authors tested the strategy on 1,411 TBI patients of the RAPID-TBI study with 2 SLMs: Phi-4-mini (3.8B parameters) and Qwen-2.5 (3B parameters). The metrics considered in the evaluation were based both on quantitative performance results (F1, sensitivity, specificity) and qualitative assessment of clinical reasoning by neurosurgery residents according to the framework QUEST. The most important findings were that CBR always worked better than classification, EBP less consistently, and that although CBR led to increased accuracy, it potentially diminished reasoning quality when too many exemplars were provided.

**Weaknesses:**

1. Low generalizability: Analysis was limited to the study of only one institution and a relatively mild TBI (81.4%) consisting of patients with GCS=15, with limited conclusions regarding the performance of the studied in other TBI severities and clinical contexts. Hence, being limited to the TBI patients reduces the ability to generalize to the field of emergency medicine in general.

2. Model behavior inconsistency: A high discrepancy between Phi and Qwen models (e.g., EBP voiding performance of Phi and increasing performance of Qwen) points to unreliability of the system and raises a question mark about system implementation readiness.

3. The studies performed in the paper are also hard to compare to the performance of a physician using the same cases, and there is no evidence of clinical value. The number of human participants involved in the evaluation is only three neurosurgery residents.

4. The findings from EBP look ambiguous due to the inconsistent and non-homogeneous implications of EBP in models, which imply that the strategy is not yet reliable to deploy clinically.

5.  The finding that improved accuracy (with more CBR exemplars) correlates with worse reasoning quality presents a concerning dilemma for clinical practice, where interpretability is crucial.

---

### Official Review · Reviewer_X468 · 2025-07-17
**Good work and very interesting**

**Confidence:** 3
**Clarity Of Writing:** good
**Clinical Significance:** great
**Methodological Novelty:** great
**Overall Rating:** 8
**Final Rating:** 8

**Experiments And Results:**

good

**Questions For The Authors:**

In comparison with LLMs how did these two SLM models performed like overall?

**Strengths:**

Very unique approach because of Dual retrieval personalization with combine of EBP and CBR. because most retrieval argument generation system focus on one form of retrieval. This research focus on ED disposition for TBI using SLMs.
Diagrams are very well done and easy to understand.
Methodology and result sections are very well documented

**Summary Of The Paper:**

This paper proposes a hybrid retrieval-augmented framework that uses small language models to unify EBP and case-based reasoning for personalized ED disposition prediction. Framework combines EBP and CBR. This framework uses two open source SLMs : Phi-4-mini and Qwen- 2.5 and multimodal dataset from the RAPID-TBI study. CBR consistently improved classification performance by increasing sensitivity and  EBP shows variable performance across models. But  its combination with CBR in Phi- 4-mini shifts predictions toward more conservative behavior.

**Weaknesses:**

Although this paper addresses why they use SLM not LLM. they could've included LLM model performance in comparison with their SLM models in the result section. Overall Bice paper, just a thought, no major weakness at all.

---

### Official Review · Reviewer_Z4nK · 2025-07-21
**Strong Accept with Minor Revision**

**Confidence:** 4
**Clarity Of Writing:** good
**Clinical Significance:** great
**Methodological Novelty:** excellent
**Overall Rating:** 8

**Experiments And Results:**

great

**Questions For The Authors:**

This work has good potential. However, I have the following questions:
1.	How was the execution time measured in Fig. 3?
2.	The paper mentions RAPID-TBI (in review) framework, however there is no details on the framework.
3.	The writing is generally well written. But there are a few grammatical inconsistences.

**Strengths:**

Identify the promising aspects of the work.
1.	This paper proposes a method to combine RAG and case-based prompting for better model performance.
2.	This study has potential to help in clinical decision making.

**Summary Of The Paper:**

The paper presents a hybrid RAG-based framework that uses both EBP and CBR in two SLMs to predict ED disposition for TBI patients. The findings suggest that combining EBP and CBR results in balanced performance in Phi-4-mini. Overall, this paper highlights the potential of combining patient-specific and evidence-guided prompting to improve the clinical utility of SLM-based decision support in emergency care.

**Weaknesses:**

The work has no major weakness.